# Dynamics of Axl Receptor Shedding in Hepatocellular Carcinoma and Its Implication for Theranostics

**DOI:** 10.3390/ijms19124111

**Published:** 2018-12-18

**Authors:** Elisa Holstein, Mathias Binder, Wolfgang Mikulits

**Affiliations:** Department of Medicine I, Division: Institute of Cancer Research, Comprehensive Cancer Center, Medical University of Vienna, 1090 Vienna, Austria; elisa.holstein@meduniwien.ac.at (E.H.); binder.mathias@imc-krems.eu (M.B.)

**Keywords:** hepatocellular carcinoma, Axl, soluble Axl, Gas6, shedding, theranostics

## Abstract

Signaling of the receptor tyrosine kinase Axl and its ligand Gas6 is crucially involved in the development of liver fibrosis and hepatocellular carcinoma (HCC) by activation of hepatic stellate cells and modulation of hepatocyte differentiation. Shedding of Axl’s ectodomain leads to the release of soluble Axl (sAxl), which is increased in advanced fibrosis and in early-to-late stage HCC in the presence and absence of cirrhosis. Here, we focus on the dynamics of Axl receptor shedding and delineate possible scenarios how Axl signaling might act as driver of fibrosis progression and HCC development. Based on experimental and clinical data, we discuss the consequences of modifying Axl signaling by sAxl cleavage, as well as cellular strategies to escape from antagonizing effects of Axl shedding by the involvement of the hepatic microenvironment. We emphasize a correlation between free Gas6 and free sAxl levels favoring abundant Gas6/Axl signaling in advanced fibrosis and HCC. The raised scenario provides a solid basis for theranostics allowing the use of sAxl as an accurate diagnostic biomarker of liver cirrhosis and HCC, as well as Axl receptor signaling for therapeutic intervention in stratified HCC patients.

## 1. Key Events in HCC

Hepatocellular carcinoma (HCC) represents the predominant type of primary liver malignancy showing the sixth most frequent incidence globally and the third most abundant cause of cancer mortality [1,2]. About 80% of HCC develop in the background of liver fibrosis and cirrhosis caused by chronic infection with hepatitis B or C virus (HBV, HCV), alcohol abuse or non-alcoholic fatty liver disease (NAFLD)/non-alcoholic steatohepatitis (NASH) [3]. Approximately 5–10% of patients in Europe are diagnosed with HCC at the Barcelona Clinic Liver Cancer (BCLC) stage 0, referring to the very early stage of HCC, allowing curative therapy such as surgical resection or orthotopic liver transplantation [4,5]. Yet, the majority of HCC patients are diagnosed at advanced stages, which limits therapeutic options to treatment with the multikinase inhibitors sorafenib or regorafenib [6].

Deregulations of oncogenes and tumor suppressor genes by mutations and altered expression have been identified, although addictions to oncogenes for targeted therapeutic intervention or tumor suppressors by synthetic lethality approaches are still missing in HCC. Major driver mutations in HCC were identified in *CTNNB1* (β-catenin), telomerase reverse transcriptase, and *TP53*, among others, leading to aberrant Wnt signaling, chromosomal instability and escape from cell death [7,8]. Mutations of *ARID1A, ARID2* and *MLL1–4* further drive HCC by affecting chromatin regulation [9,10,11]. Moreover, transforming growth factor (TGF)-β is a crucial factor in the development of HCC due to its pro-fibrotic and tumor-progressive functions [12,13]. Moreover, modulations of gene expression affect major regulators of HCC, such as c-myc, cyclin A2, cyclin D1, retinoblastoma 1, Axin1, insulin-like growth factor-II receptor/mannose-6-phosphate receptor, p16^INK4a^, Yes-associated protein (YAP)1, E-cadherin, suppressors of cytokine signaling (SOCS), interleukin (IL)-6, phosphatase and tensin homolog, or cyclooxygenase 2 in HCC [14,15]. Notably, the expression of a large panel of receptor tyrosine kinases (RTKs) is increased, including ErbB receptors, fibroblast growth factor receptors, Met and its ligand hepatocyte growth factor, vascular endothelial growth factor receptors (VEGFRs), platelet-derived growth factor receptors (PDGFRs) and TAM (Tyro3, Axl and Mer) receptors [16,17,18,19,20,21,22,23,24,25,26]. Under physiological conditions, Axl is predominantly expressed in liver endothelial cells and mostly involved in platelet aggregation and vessel integrity. However, Axl is strongly expressed in malignant hepatocytes of about 40% of HCC patients showing progression towards metastasis [27,28].

## 2. Biology of Gas6/Axl in the Liver

### 2.1. Receptor and Ligands

Together with Tyro3 and Mer, Axl belongs to the TAM receptor family of RTKs, which represents single-pass transmembrane proteins with tyrosine kinase activity. The TAM receptors share the conserved KW(I/L)A(I/L)ES sequence within the intracellular kinase domain which is unique for this family [29]. The ectodomain of Axl contains conserved tandem immunoglobulin-like domains, facilitating ligand binding, and fibronectin type III repeats (Figure 1A) [30,31]. The growth arrest specific gene (Gas)6 and Protein S (ProS) are vitamin K-dependent ligands of which Gas6 has the highest affinity for Axl, while ProS mainly binds to Tyro3 and Mer [32,33,34,35]. Both ligands share the C-terminal sex hormone-binding globulin region consisting of two laminin G-like domains and the N-terminal lipid phosphatidylserine-binding Gla domain, which is required for potentiating the TAM receptor activation, and which plays an important role in phagocytosis [36,37]. TAM receptor activation requires vitamin K-dependent γ-carboxylation of Gas6 and ProS [38,39]. Gas6 binds to Axl in a 1:1 stoichiometry due to the binding of the two immunoglobulin-like domains of Axl to the laminin G-like domain of Gas6 (Figure 1B). This leads to homo-dimerization of Axl without direct contacts neither between the Axl nor between the Gas6 monomers [31,33]. Additionally, the tubby-like protein-1 is able to interact with TAM receptors, while tubby and Galectin-3 only activate Mer [40,41]. Ligand binding leads to trans-autophosphorylation of the intracellular domain (ICD) tyrosine residues, which provide docking sites for recruitment of intracellular signaling effectors [29]. Interestingly, overexpression of TAM RTKs could result in ligand-independent activation [29,30,42].

Axl-mediated proliferation and survival depend on the mitogen activated protein kinase (MAPK)/extracellular-signal regulated kinase (ERK) pathway and involve phosphoinositide 3-kinase (PI3K) and c-Jun-N-terminal kinase (JNK) activation [43,44]. Additionally, Ras, Twist and nuclear factor κ-light-chain-enhancer of activated B cells (NF-κB) are downstream targets of Axl [29,45,46]. Yet, the Axl/PI3K/Akt and the Mer/PI3K/Akt pathways differ, since NF-κB is upregulated upon Axl activation, while Mer signaling downregulates NF-κB [47,48]. Moreover, Axl is targeted by YAP1 and a mediator of YAP-dependent oncogenic signaling via ERK1/2 [49]. Furthermore, microenvironmental factors such as hypoxia activate Axl signaling in order to drive angiogenesis [50]. The hypoxia-inducible factor (HIF)-1α and HIF-2α bind to the hypoxia-response element in the Axl proximal promotor leading to transcriptional activation of Axl in HCC [51]. Another upstream regulator of Axl is RAB10, which is associated with increased tumor size and advanced tumor staging in HCC patients [52].

Axl was originally found as a transforming gene in leukemia cells and has the ability to transform fibroblasts and myeloid cells through overexpression [53,54,55]. Axl is cleaved by a disintegrin and metalloproteinase (ADAM)10 and ADAM17 in a protein kinase C (PKC)-dependent fashion causing the release of soluble Axl (sAxl) which maintains the ability to interact with Gas6 [54,56,57]. Thus, the release of sAxl and its involvement in a negative feedback loop by Gas6 binding together with the γ-secretase-mediated release of a soluble Axl ICD suggests bidirectional signaling as shown for ErbB4 [58]. Since sAxl is able to bind Gas6 and therefore capable of depleting the ligand, it is considered to be a critical determinant in affecting autocrine or paracrine Axl signaling [57]. In addition to the ligand depletion by sAxl, the ICD of Axl could remain active supporting the idea of Gas6-independent signaling. In contrast, it has also been hypothesized that ectodomain shedding of Axl is a mechanism of dampening the signal cascade [59]. 

### 2.2. Dichotomic Role of Axl in Cancer

Dysregulation of Axl signaling causes inflammation, autoimmune disease and cancer [33]. Moreover, endogenous or exogenous overexpression of Axl alone can be sufficient to induce oncogenesis, emphasizing the transforming activity of Axl [33,55]. In the context of cancer, TAM RTKs and particularly Gas6/Axl signaling contribute to survival of tumor cells in response to apoptotic stimuli [33,60]. The Axl ligand Gas6 is secreted by tumor cells, the vasculature, tumor-infiltrating leukocytes and bone marrow progenitor cells in the tumor microenvironment (TME) [27,61]. Tumor-associated macrophages are stimulated to increase their Gas6 expression in the TME in comparison to resident tissue macrophages, which implicates a positive feedback loop in Gas6/Axl signaling [61]. In HCC, the Gas6/Axl pathway has been identified to promote tumor invasion by activating Slug [62]. Increased activation of Axl signaling mediates invasion and metastasis in various cancers, such as breast, lung, prostate and pancreatic cancer [63,64,65,66,67]. In line, Axl was identified as a crucial regulator of epithelial to mesenchymal transition (EMT) and chemoresistance (Figure 2) [68,69]. In HCC, malignant hepatocytes undergo EMT due to the cooperation of TGF-β with Axl signaling, which leads to invasion of HCC cells [28]. The interaction of Gas6/Axl with 14-3-3ζ activates JNK which subsequently switches TGF-β towards tumor-progressive functions by aberrant phosphorylation of the Smad3 linker region and increased transcription of pro-metastatic genes. Additionally, Axl regulates angiogenesis in tumors, which is indicated by impaired endothelial tube formation upon inhibition of Axl signaling [70,71]. 

TAM RTK signaling leads to metastasis as Axl and Mer inhibit the anti-tumor response of natural killer (NK) cells which are consequently disabled to kill disseminating cancer cells [72]. Beyond that, Gas6/Axl signaling activates hepatic stellate cells (HSCs) to transit into myofibroblasts which results in fibrosis [73]. Additionally, Gas6/Mer signaling is suggested to be involved in liver fibrogenesis [74]. As described, liver fibrosis predisposes patients for HCC, which indicates that Gas6/Axl represents an oncogenic driver. Notably, Gas6/Axl is activated on aggregating platelets in the presence of phosphatidylserine, which allows stabilization of clot formation under physiological conditions [33,75]. After activation of PI3K/Akt, the cytoplasmic tail of β3 integrin is phosphorylated leading to outside-in signaling [76]. This mediates a change in platelet shape, clot retraction and thrombus stabilization [75]. In this line, thrombosis is prevented in Gas6 knock-out (KO) mice due to impaired platelet aggregation [77]. Moreover, we hypothesize that TAM signaling might be involved in cancer-associated thrombosis since platelets in general are able to protect tumor cells in the circulation in order to facilitate metastatic colonization at distant sites [78]. Paracrine signaling of platelets suppresses the NK cell mediated lysis, which promotes tumor cell survival [79,80]. Thus, cancer patients are at high risk of developing deep vein thrombosis and pulmonary embolisms due to elevated expression and activity of Axl [81]. 

Axl is involved in chemoresistance since it is overexpressed in, for example, imatinib resistant tumor cells [82]. Inhibition of Axl by miRNA-34a-5p decreases chemoresistance of HCC cells to cisplatin [83], which was comparably observed in breast cancer [84]. In acute myeloid leukemia, increased Axl expression is associated with resistance against the chemotherapeutics doxorubicin, VP16 and cisplatin, as well as to FLT3 inhibitors [85,86]. Axl can also heterodimerize, for instance, with EGFR or ErbB2, which leads to activation of alternative signaling routes, also called bypass signaling, and diversification of downstream signaling facilitating chemoresistance [69,87,88,89]. On the one hand, inhibition of MAPK signaling (MAPKi) results in increased transcription of multiple RTKs via bypass signaling [90]. On the other hand, MAPKi leads to reduced ectodomain shedding as shown for Axl. Increased Axl expression further supports bypass signaling by increased JNK phosphorylation. Therefore, ectodomain shedding of Axl is a feedback mechanism that mediates the compensatory bypass signaling of JNK after MAPKi. Increased ectodomain shedding is able to inhibit this exit route by blocking bypass signaling and, thus, to antagonize TAM-mediated mechanisms of chemoresistance [90]. 

Contrary to oncogenic traits, TAM RTKs are important in the process of phagocytosis and suppression of the inflammatory cytokine response [33,91]. In particular, the clearance of apoptotic cells is TAM-dependent and plays a major role in the resolution of inflammation. Therefore, one of the physiological functions of the TAM family is to provide an inhibitory feedback mechanism, which promotes tissue repair [92,93]. In particular, Axl and Mer may play a major role in the anti-inflammatory process since they prevent and terminate innate immune-mediated inflammation [94]. The loss of all three TAM receptors results in autoimmunity, hyper-inflammation, hyperproliferation of lymphocytes and hepatitis [95,96]. In order to facilitate a proper immune response, Toll-like receptors (TLRs) regulate TAM signaling upstream to prevent chronic activation of antigen-presenting cells [35]. Activation of TLRs releases pro-inflammatory cytokines resulting in activation of type I interferon (IFN) receptors (IFNAR). IFNAR signaling through Janus kinase/signal transducers and activators of transcription (JAK/STAT) induces expression of the TAM RTKs, which then shut down the immune response via SOCS1, SOCS3, NF-κB inhibition and repression of the IFN regulatory factor 3 [94,97]. Inflammatory cytokines such as IL-1β, IL-6, tumor necrosis factor (TNF)-α, IFN-α and -β are upregulated in the liver of TAM triple KO mice [96]. This TAM-induced feedback mechanism facilitates effective acute inflammation while preventing chronic disease. Since chronic inflammation is considered to be a major driver of tumorigenesis, TAM signaling acts tumor-suppressive due to its anti-inflammatory role (Figure 2) [35,98]. Furthermore, Gas6/Axl is suggested to positively regulate the expression of the tumor suppressor LIGHT (lymphotoxin-related inducible ligand that competes for glycoprotein D binding to herpesvirus entry mediator (HVEM) on T cells mediating an anti-oncogenic trait) which induces robust anti-tumor immunity [99]. Although tumor-associated macrophages are suggested to secrete Gas6 in the TME [100], the role of Gas6/Axl for the infiltration of macrophages remains an open issue. There might be a continuum of classically activated macrophages promoting acute inflammation (M1) and alternatively activated macrophages suppressing inflammation while promoting tissue repair and tumor progression (M2), leading to opposing phenotypes. 

### 2.3. Role of Gas6/Axl under Healthy and Pathological Conditions in the Liver

Axl causes resistance to TGF-β induced growth inhibition and decreases the susceptibility to TGF-β-induced apoptosis in HCC cells [28]. Furthermore, Axl elicits abilities for invasion and trans-endothelial migration of EMT-transformed HCC cells [26,28]. PI3K/Akt and PAK1 signaling is strongly reduced in Axl KO, as inhibition of these pathways leads to decreased HCC cell migration and invasion [26]. Axl KO results in reduced liver fibrosis in vivo following CCl4 treatment [73]. HSCs cultured from Axl KO mice exhibited decreased expression of markers for HSC activation, such as smooth muscle actin and collagen type 1, indicating reduced HSC activation.

Under healthy conditions, Gas6 is not secreted by HSCs, yet Kupffer cells contribute to Gas6 expression in the normal liver. Upon CCl4-induced liver injury, Gas6 secretion is stimulated in HSCs and in infiltrating macrophages [101]. Gas6 plays an important role in steatohepatitis and progression to liver fibrosis as Gas6 KO mice show a decreased disease burden. Furthermore, hepatic inflammation is reduced by limited infiltration of F4/80-positive macrophages and by decreased expression levels of IL-1β, TNF-α, monocyte chemotactic protein (MCP)-1 and lymphotoxin-β in Gas6-deficient mice. Attenuated myofibroblast activation and fibrogenesis together with lower TGF-β and collagen type 1 expression is detected in Gas6-deficient mice subjected to a choline-deficient, ethionine-supplemented diet. In both models, Gas6 deficiency limits macrophage recruitment leading to a decreased amount of inflammatory cytokines in the liver [102]. Furthermore, Gas6 deficiency leads to delayed and limited liver regeneration in the CCl4-injured liver. The levels of MCP-1 secreted by activated Kupffer cells are reduced in Gas6 KO mice along with decreased IL-6, TNF-α and CD14 levels. These findings are consistent with defective Kupffer cell activation resulting in delayed and limited liver repair [103]. 

Gas6 is further involved in hepatic graft versus host disease (GVHD) since apoptosis of liver cells in portal spaces is significantly lower in Gas6 KO mice. This indicates that Gas6 deficiency reduces liver GVHD in recipients of allogenic bone marrow transplantation [104]. In a murine model of partial hepatic ischemia/reperfusion (I/R) injury, Gas6 is required for resolving partial I/R due to increased cell survival in the hepatic parenchyma. Furthermore, the levels of inflammatory mediators such as TNF-α and IL-1β are significantly elevated in the Gas6-deficient mice exposed to I/R [105].

In HCC, EMT-transformed hepatocytes rather than differentiated hepatocytes upregulate the expression of Axl and secrete Gas6, suggesting autocrine regulation of Gas6/Axl signaling (Figure 3) [28]. In the background of fibrosis, sinusoidal endothelial cells, activated HSCs and Axl-positive- myofibroblasts, as well as Kupffer cells, release Gas6 into the TME of HCC, causing a Gas6-enriched HCC stroma. These data suggest that Axl signaling drives HCC progression in the presence of large amounts of bioactive Gas6. 

## 3. Cell Communication Regulated by Ectodomain Shedding

Proteolytic cleavage of extracellular domains of transmembrane proteins is mainly performed by sheddases. This process referred to as ectodomain shedding is a posttranslational mechanism critically required for generating and regulating the active form of multiple growth factors, membrane-bound precursor ligands, adhesion molecules, cell surface receptors and ligand-independent signaling. All these functions of shedding facilitate signaling on host and neighboring cells [106,107]. Its regulatory function essentially affects cell–cell communication in development and tissue homeostasis, which is underlined by the observation that, e.g., deletion of ADAM17, one of the most prominent sheddases, is embryonic lethal [108]. Proteolytic cleavage of transmembrane proteins by sheddases such as ADAMs and matrix metalloproteases (MMPs) is induced by certain stimuli involving phorbol esters, calcium ionophores and cytokines such as TNF-α, IFN-γ, IL-2, VEGF or IL-1β [109,110,111,112,113,114]. The subsequent activation of signaling pathways such as MAPK-, PI3K/Akt/mammalian target of rapamycin-, PKC-, as well as calcium-dependent signaling pathways cause ectodomain shedding [90,115].

Shedding of pro-TGF-α by ADAM17 drives tumorigenesis by the release of soluble TGF-α and activation of the epidermal growth factor receptor (EGFR) signaling [116,117]. ADAM17, also known as TNF-α converting enzyme (TACE), was initially found to be the protease cleaving TNF-α. Over time, a large number of ADAM17 target molecules were identified, among them ligands for EGFR, L-selectin, CD44 and VEGFR. All EGFR ligands are synthesized as membrane-bound precursors, which require ectodomain shedding for releasing the active ligand [118]. For instance, the membrane-bound precursor pro-heparin-binding EGF-like growth factor (HB-EGF) is shedded by several ADAMs [119]. Both, soluble HB-EGF and pro-HB-EGF are biologically active. The precursor molecule is able to act on neighboring cells in a juxtacrine manner as it is membrane-anchored, whereas the soluble protein can diffuse to distant sites leading to EGFR activation. Additionally, the C-terminal fragment of HB-EGF is internalized after proteolytic cleavage and able to transcriptionally modulate target genes in the nucleus. In this way, HB-EGF signals bidirectionally as the soluble ectodomain of HB-EGF binds to and activates EGFR, while the soluble ICD acts on transcriptional repressors causing activation of target genes [120,121]. Similarly, γ-secretase cleavage after ectodomain shedding occurs in ErbB4/HER4 leading to nuclear translocation of the soluble ICD [122]. In addition, the ErbB4-ICD keeps its tyrosine kinase function, while the ectodomain of ErbB4 is able to deplete the ligand causing a negative feedback loop [123]. Signaling activities of both ErbB4 cleavage products demonstrate bidirectional signaling as described for HB-EGF. 

In line, the RTK Axl undergoes a γ-secretase-dependent cleavage after ectodomain shedding generating a soluble ICD [58]. A comparable mechanism to ErbB4 could be involved in Axl signaling since sAxl as well as Axl-ICD show biological activity resulting in ligand depletion by sAxl and target gene activation by Axl-ICD [57,124]. Further studies need to clarify whether Axl-ICD is able to maintain its tyrosine kinase activity like ErbB4-ICD, which would provide more information on ligand-independent signaling of Axl. Yet, target gene activation is not the only outcome of γ-secretase cleavage, as the soluble ICD can also be rapidly degraded, which is shown for Met and Tie1 [125,126]. Degradation of ICDs is an antagonistic mechanism to ligand-independent signaling. Therefore, γ-secretase cleavage of Axl can either lead to targeted degradation or to nuclear translocation of Axl-ICD, resulting in the cease of Axl signaling or activation of target genes. 

Of note, the extracellular domain of ErbB2 is shedded by ADAMs and MMPs upon receptor activation. Notably, the resulting serum-detectable soluble fragment is able to reflect the activity state of the receptor since the ICD of ErbB2 is shown to be constitutively active [127,128]. An engineered deletion of the extracellular domain of ErbB2 increases the oncogenic potential through increased kinase activity due to ligand-independent signaling [129]. In this context, it remains an open issue whether ErbB2-like ligand-independent signaling is relevant for Axl functions. 

## 4. Axl Receptor Shedding: Gas6-Dependent Signaling versus Signal Dampening

A previous study by Ekman et al. showed that Gas6 is bound to sAxl in human serum and plasma by an excess of sAxl [57]. The authors concluded that free Gas6 is captured by sAxl which suggests that sAxl downregulates Axl signaling by ligand depletion [57]. In consequence, HCC progression should be subsequently attenuated by diminished Gas6/Axl signaling. However, serum Gas6 levels are increased in patients with advanced fibrosis and cirrhosis as well as in HCC patients, suggesting that excess Gas6 is able to overcome the feedback inhibition of sAxl [73,130], albeit, these data have to be confirmed in larger patient cohorts. In HCC, there is not only a positive correlation of serum Gas6 and sAxl levels with increased tumor staging, but there might be even an excess of Gas6 levels compared to sAxl [131]. Additionally, there is also increased expression of ADAM10 in HCC which is associated with tumor progression [132,133]. Increased ADAM10 expression might be involved in mediating increased Axl ectodomain shedding. As Gas6 is secreted by various liver cell types in the TME (Figure 3), abundant levels of free Gas6 might be available to bind cognate Axl receptors and to activate Axl signaling for cancer progression [61]. In line, Axl expression is upregulated in HCC and increased sAxl levels were found in patients with liver cirrhosis as well as very early to late stages of HCC [134,135]. 

In this scenario, it is also conceivable that Axl transduces an anti-oncogenic signal, which is dampened by sAxl leading to an oncogenic phenotype (Figure 4A). Yet, Gas6/Axl signaling induces the expression of pro-metastatic genes, such as *SNAI1*, *MMP9* and *TGF-β1* in HCC, which are crucially involved in cell invasion and trans-endothelial migration of EMT-transformed hepatocytes [12,28]. Moreover, high Axl expression as well as high sAxl levels independently correlate with poor HCC patient survival, indicating that Gas6/Axl expression associates with a tumor-promoting rather than a tumor-suppressive signaling [28,134]. Consequently, these findings are contradictory to the hypothesis that ectodomain shedding of Axl leads to signal dampening. If sAxl is able to inhibit Gas6/Axl signaling, the oncogenic function should be blocked or at least attenuated. In this case, higher sAxl concentrations would implicate decreased disease progression and prolonged survival of HCC patients, which is not conclusive with recent data [134,135]. Since cancer patients with increased Axl expression show disease progression and advanced tumor staging resulting in rather metastatic phenotypes [27], the hypothesis is supported that Gas6 levels are in excess compared to sAxl and free Gas6 is available to activate non-shedded Axl receptor for amplifying Gas6-dependent Axl signaling in liver fibrosis and HCC (Figure 4B, right panel).

Recent findings support the idea of Gas6-dependent Axl signaling in liver fibrosis which has been revealed by Gas6 KO and Axl KO studies, showing reduced fibrogenesis [73,102]. In addition, latest studies show that sAxl levels are increased in advanced fibrosis/cirrhosis [73,135]. Notably, if increased sAxl levels deplete Gas6, Gas6 deficiency will not affect the development and progression of fibrosis. Since sAxl levels are increased in fibrosis, whereas either Gas6 or Axl deletion reduces fibrosis, we conclude that Gas6/Axl signaling is pro-fibrotic and that Gas6 levels exceed those of sAxl. Regarding the hypothesis of signal dampening, excess of sAxl should decrease fibrogenesis through capturing free Gas6, resulting in a phenotype comparable to the one of chemically challenged Gas6 KO mice [102]. In conclusion, observations from studies of Gas6 KO mice are in accordance with recent findings in patients and underline Gas6-dependent Axl signaling in liver fibrosis. Yet, solid evidence for the oncogenic Gas6-dependent Axl signaling in HCC requires further studies in HCC patients by determining levels of sAxl and free Gas6 as well as those levels of Gas6/sAxl complexes in patient blood. 

If the Gas6 levels exceed the sAxl values, it will indicate that Gas6 overcomes the inhibitory feedback mechanism. This scenario suggests that there might be a switch predisposing fibrosis, cirrhosis or HCC development, where the inhibitory Axl shedding mechanism is circumvented. In the case that serum Gas6 is lower than sAxl, the Gas6/Axl signaling would be dampened, which would rather lead to the abrogation of an anti-fibrotic or anti-oncogenic signal driving fibrosis or HCC. We propose that cleavage of Axl does not contribute to the development or progression of liver fibrosis/cirrhosis and HCC due to the presence of abundant non-shedded Axl receptors overcoming the loss of proteolytically cleaved Axl. Available free Gas6 is then able to bind increasingly expressed Axl receptor and stimulate Gas6/Axl signaling driving HCC progression (Figure 4B, right panel). 

The Gas6-independent signaling hypothesis implicates that proteases are recruited to cleave the Axl ectodomain after Gas6-mediated Axl activation (Figure 4B, left panel). In this scenario, the ICD could remain active and could be still able to phosphorylate effector molecules. As a result, high Axl receptor expression would lead to increased Gas6-independent signaling, promote tumor growth as well as EMT and metastasis. However, it is an open question whether ectodomain shedding occurs after Axl homo-dimerization and ICD activation. Interestingly, the mechanism of shedding prior to receptor activation and ligand-independent signaling has been reported for ErbB2 [128]. 

## 5. Theranostics: Diagnostic and Therapeutic Potential of Gas6/Axl

Patients with early diagnosed HCC achieve a five-year survival rate of 60% after liver resection and a five-year survival rate of 67% post transplantation [136]. Unfortunately, the majority of HCC patients is diagnosed at advanced stages, resulting in a survival median of less than one year [137]. The question is whether the analysis of Gas6/sAxl levels in patients’ blood can be used to diagnose advanced fibrosis/cirrhosis as well as HCC. Interestingly, recent large-scale multicenter studies demonstrated that sera of all HCC patients, including very early and early HCC, show elevated sAxl levels in the presence as well as absence of cirrhosis [134,135]. Receiver operating characteristic curve analysis of sAxl indicated an increased sensitivity and specificity to detect cirrhosis and HCC superior to α-fetoprotein (AFP), which is used in HCC diagnostics. In addition, the combination of sAxl with AFP further increases the diagnostic accuracy in very early to late HCC patients [134]. As no false positive sAxl signals could be detected in, e.g., chronic HBV or HCV infection or NAFLD/NASH patients or secondary liver malignancies from colorectal cancer, sAxl is the most promising novel serum biomarker for the accurate detection of HCC. Yet, serum sAxl levels are also increased in chronic degenerative diseases of other organs such as the heart which could reduce the diagnostic specificity of sAxl [138].

With respect to the therapeutic intervention, the Gas6/Axl signaling pathway represents a promising target in HCC. For instance, an engineered Axl decoy receptor that was shown to trap Gas6 with higher affinity than the wildtype Axl receptor could lead to effective sequestration of Gas6 and thus, to a decrease of Axl signaling [139]. Targeting Axl signaling with this Axl decoy receptor has shown promising results in models of ovarian and breast cancer metastasis. In contrast to the anti-Axl tyrosine kinase inhibitors (TKis) foretinib and BGB324, the Axl decoy receptor specifically inhibits Axl signaling without affecting Tyro3 or Mer activity. Despite comparable efficacies of Axl decoy receptor and TKis, foretinib caused side effects in 91% of patients in a phase II trial [140]. Nevertheless, it has not been clarified whether the Axl decoy receptor affects the immune system or if there are adverse effects on inflammation. In addition, the synergy of Axl and TGF-β leading to neutrophil infiltration renders an alternative target for HCC therapy involving the TME [141]. Interestingly, the secretion of the neutrophil attractant CXCL5 depends on the molecular collaboration of Gas6/Axl and TGF-β-signaling in TGF-β-positive HCC patients [141]. CXCL5 causes chemotaxis and activation of neutrophils, which leads to intra-tumoral macrophage and T cell infiltration resulting in enhanced HCC progression [142]. CXCL5 expression could be directly targeted by pharmacological intervention in order to avoid harmful effects of either anti-TGF-β or anti-Axl therapies. 

## 6. Conclusions

Ectodomain shedding is a posttranslational mechanism that either acts through the released extracellular domain, the soluble ICD, or bidirectionally through both. Proteolytic cleavage of Axl has not been sufficiently studied yet, which raises multiple questions. In particular, the role of Gas6/Axl signaling in liver fibrosis and HCC requires further elucidation in order to test several hypotheses. As suggested by Ekman et al., ectodomain shedding of Axl could serve as an inhibitory feedback mechanism to shut down the Gas6/Axl signaling route [57]. In this hypothesis, Gas6/Axl signaling is dampened by sAxl. As shown for ErbB4, the soluble extracellular domain could be able to neutralize the ligand, which would support Ekman et al.’s hypothesis. Nevertheless, γ-secretase activity after ectodomain cleavage could still mediate signaling to the nucleus which would result in bidirectional signaling [57,124]. An alternative hypothesis suggests that abundant free Gas6 in the TME could overcome this inhibitory feedback mechanism for promoting fibrosis and HCC. Additionally, shedding of activated Axl receptors could lead to Gas6-independent signaling driving disease progression. 

In conclusion, the emerging role of Axl in theranostics of HCC is reflected by the promising results of sAxl as an innovative diagnostic biomarker. Beyond, sAxl levels are able to stratify patients according to liver disease stages indicating that Axl as well as sAxl are valuable tools for precision medicine. In addition, CXCL5 might be a promising therapeutic target in Axl/TGF-β-positive patients. To further elucidate the role and mechanism of Gas6/Axl signaling, the Gas6 as well as the Gas6/sAxl complex levels have to be analyzed in blood samples of fibrosis and HCC patients. Examining the combination of Gas6 and Axl will clarify the role and mechanism of ectodomain shedding and might strengthen diagnostic and therapeutic strategies to combat liver fibrosis and HCC.

## Figures and Tables

**Figure 1 ijms-19-04111-f001:**
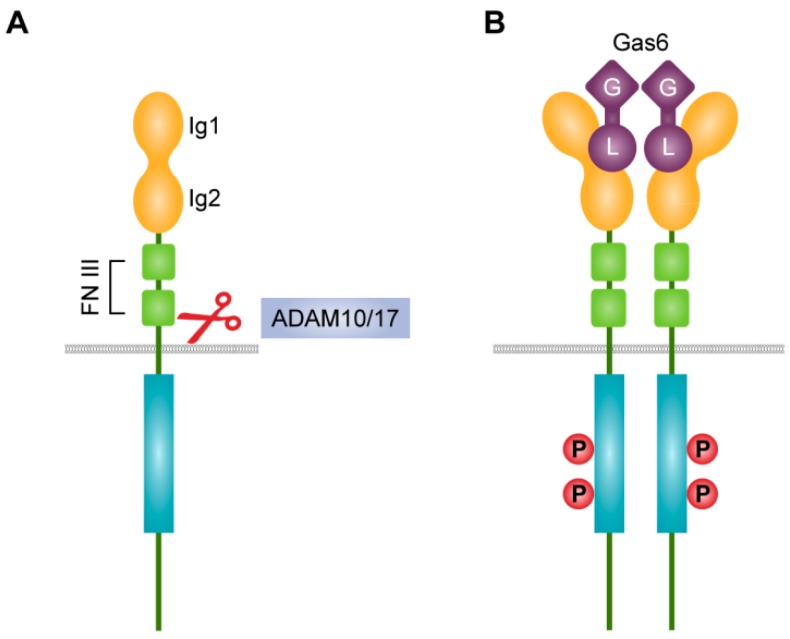
Structure and activation of Axl receptor. (**A**) The ectodomain of Axl consists of the N-terminal immunoglobulin-like Ig (Ig1, Ig2) and two fibronectin (FN) type III domains. ADAM10 and ADAM17 can cleave the ectodomain between valine 438 and tryptophan 452 (V438-W452) close to the transmembrane region. Red scissors indicate proteolytic cleavage. (**B**) Gas6/Axl homo-dimerizes in a 1:1 stoichiometry resulting in the tyrosine phosphorylation of ICDs. Gas6 binds to the Ig domains of Axl by the carboxy-terminal laminin G-like domain (L). The amino-terminal Gla domain of Gas6 (G) binds to the lipid phosphatidylserine.

**Figure 2 ijms-19-04111-f002:**
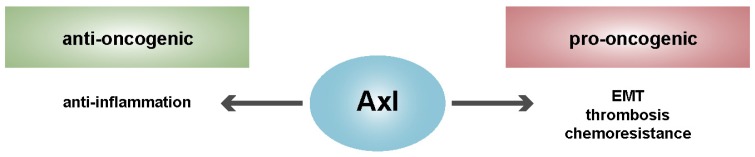
Dichotomy of Axl by anti- and pro-oncogenic actions. Axl prevents chronic inflammation while causing changes in epithelial cell plasticity, aggregation of platelets and escape from chemosensitivity. EMT, epithelial to mesenchymal transition.

**Figure 3 ijms-19-04111-f003:**
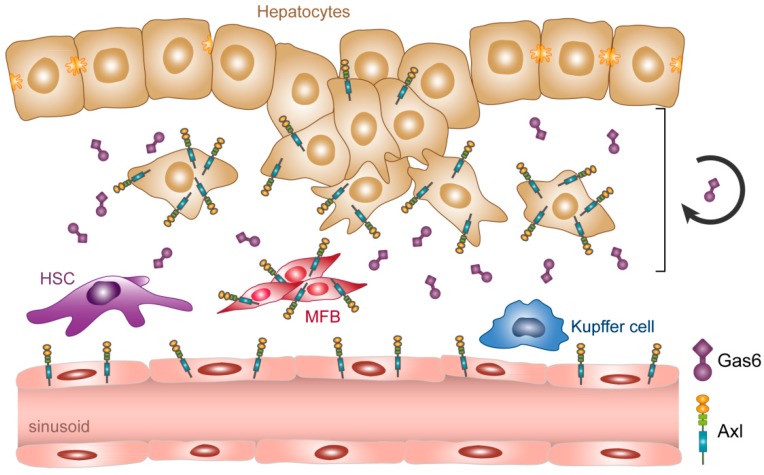
Expression of Gas6 and Axl in hepatocellular carcinoma (HCC). Malignant hepatocytes undergoing de-differentiation by EMT as well as HSCs trans-differentiating to MFBs upregulate Axl expression. EMT-transformed hepatocytes secrete Gas6 causing autocrine Gas6/Axl signaling. Activated HSCs/MFBs and Kupffer cells release Gas6 into the tumor microenvironment (TME). Circled arrow indicates autocrine Gas6/Axl signaling. HSC, hepatic stellate cell; MFB, myofibroblast.

**Figure 4 ijms-19-04111-f004:**
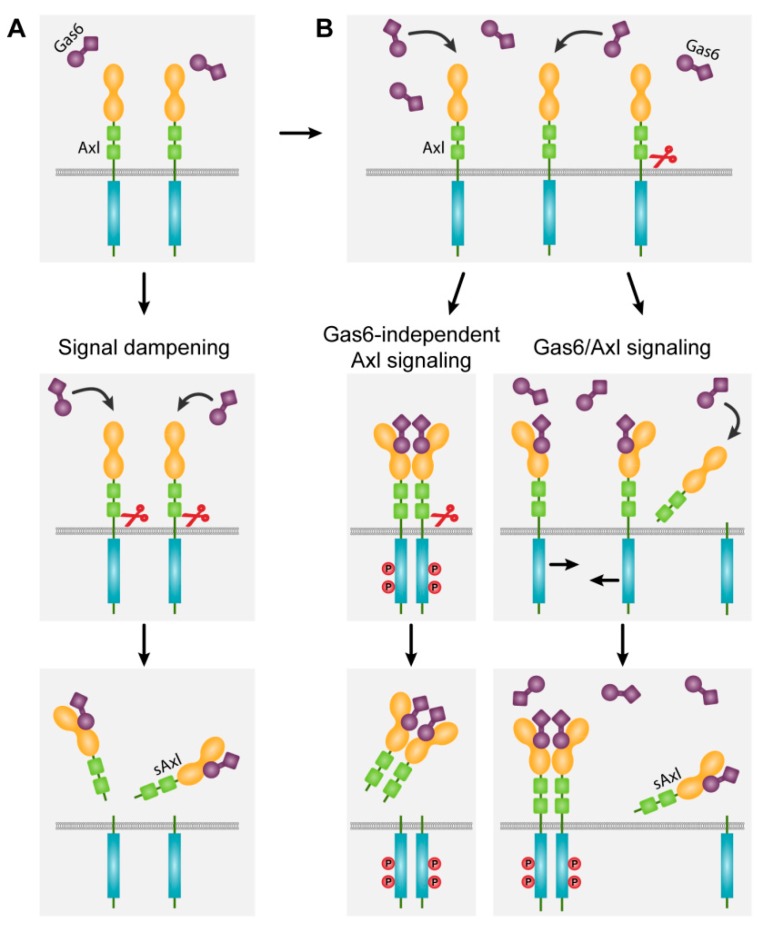
Axl signaling in liver fibrosis and HCC after ectodomain shedding. (**A**) Dampening of Gas6/Axl signaling. The monomeric Axl receptor is cleaved by proteases resulting in sAxl monomers that bind to and trap free Gas6. (**B**) Gas6-independent and Gas6-dependent Axl signaling. Middle left and lower panel: proteases cleave Gas6/Axl tetramers releasing sAxl dimers. Signaling of Axl receptor lacking the extracellular domain is questionable. Middle right and lower panel: In the presence of Gas6/sAxl complexes, abundant free Gas6 binds to Axl allowing homo-dimerization of receptors and activation of Axl signaling. Red scissors indicate proteolytic cleavage.

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
