# Peer review of "Dynamics of Axl Receptor Shedding in Hepatocellular Carcinoma and Its Implication for Theranostics"

_ijms, 2018, doi:10.3390/ijms19124111_

Reviewer 1 Report

The present manuscript provides an in-depth review of the knowledge on the Axl tyrosine kinase implication in liver disease. It suggests also possible areas of interest for future research in the field.

There are few suggestions that should be considered by the authors.

The authors focus on Axl, while they do not mention recent studies on the possible implication of MerTK in liver fibrosis. It could be of importance to include a short paragraph with this topic.

There are some minor specific points, indicated by the line where they appear.

L114-115 "...increased ectodomain shedding can be a mechanism of escape from chemosensitivity, as it is suggested for the ErbB2-associated resistance to trastuzumab [60]." Please indicate also ref. 39 in this context, as they study this effect on Axl shedding. The sentence could be rephrased for clarity, as it is confusing the concept "escape from chemosensitivity". Rather they authors could mention that reduced shedding induces chemosensitivity.

The graphic representation used for Gas6 is misleading. The authors seem to represent the binding domains of Gas6 (LG modules), rather than the full length molecule that will contain the Gla domain for phosphatydil-serine binding.

L124. The authors should consider citing the study PMID:25265470 about the ligand requirements for the TAM family. In this context, if they cite the initial studies on this subject (like ref. 64), possibly they should include PMID:7854420, as this was the first report identifiing Gas6 as the Axl ligand.

197-199. Although TAM receptors in platelets are pro-aggregant, it has not been established its function in cancer-associated thrombosis.

332 Miss-spelling “Gas6/Axl signaling”

335 Although the authors mention the specificity of high sAXL in serum of HCC, sAXL levels are increased in advanced fibrosis and cirrhotic patients. This should be made clear in the context of specificity.

Besides, several reports indicate that sAXL could correlate with chronic degenerative pathology in other organs besides the liver, and this could be decrease the diagnostic specificity of the biomarker (heart; PMID:24681018; lung; PMID:29634284).

Author Response

Notes to the comments of reviewer 1:

To comment 1:

The authors focus on Axl, while they do not mention recent studies on the possible implication of MerTK in liver fibrosis. It could be of importance to include a short paragraph with this topic.

We agree with this suggestion and included a statement on the role of MerTK in liver fibrogenesis. See chapter 3/part B, which is chapter 2/part B of the revised version.

To comment 2:

L114-115 "...increased ectodomain shedding can be a mechanism of escape from chemosensitivity, as it is suggested for the ErbB2-associated resistance to trastuzumab [60]." Please indicate also ref. 39 in this context, as they study this effect on Axl shedding. The sentence could be rephrased for clarity, as it is confusing the concept "escape from chemosensitivity". Rather they authors could mention that reduced shedding induces chemosensitivity.

We completely agree with this comment. However, we deleted the statement dealing with the involvement of ectodomain shedding on the ErbB2-associated resistance to trastuzumab since reviewer 2 suggested to revise chapter 2 (see comment 3 to reviewer 2).

To comment 3:

The graphic representation used for Gas6 is misleading. The authors seem to represent the binding domains of Gas6 (LG modules), rather than the full length molecule that will contain the Gla domain for phosphatydil-serine binding.

Thank you for this suggestion. In order to clarify the domains of Gas6, we have changed the graphic representation. The laminin G-like domain of Gas6 binding to Axl is depicted as a circle, while the Gla domain binding to lipid phosphatidylserine is represented by a rectangle. Corresponding graphic changes were introduced into Figure 1, 3 and 4 as well as into the Graphical Abstract. Abbreviations for the laminin G-like (L) and the Gla domain (G) were incorporated into the Legend to Figure1.

To comment 4:

L124. The authors should consider citing the study PMID:25265470 about the ligand requirements for the TAM family. In this context, if they cite the initial studies on this subject (like ref. 64), possibly they should include PMID:7854420, as this was the first report identifying Gas6 as the Axl ligand.

We appreciate this suggestion and cited PMID:25265470 showing the ligand requirements for the TAM family. We further included the original study (PMID:7854420) describing the Gas6 binding to Axl. See chapter 3/part A, which is chapter 2/part A of the revised version.

To comment 5:

197-199. Although TAM receptors in platelets are pro-aggregant, it has not been established its function in cancer-associated thrombosis.

As recommended by the reviewer, we changed our wording and hypothesize - due to the lack of experimental evidence - that Axl plays a crucial role in cancer-associated thrombosis (chapter 3/part B, which is chapter 2/part B of the revised version)

To comment 6:

332 Miss-spelling “Gas6/Axl signaling”

Thank you. We corrected the misspelling.

To comment 7:

335 Although the authors mention the specificity of high sAXL in serum of HCC, sAXL levels are increased in advanced fibrosis and cirrhotic patients. This should be made clear in the context of specificity.

We closely followed the suggestion of the reviewer and clarified that the cleavage of Axl does not contribute to the development or progression of the liver fibrosis/cirrhosis and HCC due to the presence of ………...

To comment 8:

Besides, several reports indicate that sAXL could correlate with chronic degenerative pathology in other organs besides the liver, and this could be decrease the diagnostic specificity of the biomarker (heart; PMID:24681018; lung; PMID:29634284).

We appreciate this comment and incorporated a notion into chapter 5 dealing with a possibly reduced diagnostic specificity of sAxl in patients suffering from chronic degenerative diseases such as the heart. Therefore, PMID:24681018 was cited. We hope for the understanding of the reviewer that we did not include PMID:29634284 as increased Gas6/Axl signaling in lung fibrosis is associated with decreased rather than increased serum sAxl levels.

Reviewer 2 Report

This manuscript described the significance of the Gas6/Axl axis by citing a sufficient number of research papers. The topic is interesting and has an originality, and the pathway is attractive for most biomedical researchers of liver diseases. However, the contents are not well organized, and comprehensively summarized. The title does not match with a part of the contents. The following points should be addressed:

1. There are a number of the mixed descriptions about Axl in HCC and other liver diseases, or in liver and other tissues. In addition, there are many redundant descriptions. The authors should clearly mention in each sentence what this description is related to. And the manuscript construction should be revised thoroughly.

For instance,

   1) General information of Axl

   2) a. physiological significance of Axl in liver

     b. pathological significance of Axl in liver fibrosis/cirrhosis

     c. pathological significance of Axl in HCC

   3) a. molecular functions of Axl in liver

     b. molecular functions of Axl in liver fibrosis/cirrhosis

     c. molecular functions of Axl in HCC

   ··· (just an example)

2. Figure 1. Please add gamma-secretase and its cleaving site.

3. The chapter 2 is not necessary because it is mainly focusing on other molecules than Axl, or should be revise by focusing on Axl.

4. How is the expression of Axl and Gas6 regulated in HCC? If such information is available, the authors should add it to the manuscript.

5. Lines 214-237. How do TAM RTKs suppress inflammation in livers? If its molecular mechanism is revealed in detailed, it should be explained. What effects do they have on hepatitis?

6. The chapter 4. The authors should discuss about serum sAxl levels in HCC patients from the viewpoint of ADAM17 and/or gamma-secretase expression.

7. Lines 299-300. This sentence is difficult to understand. I think that, even though sAxl suppresses the Gas6/Axl signaling, the excess amount of Gas6 compared to sAxl can explain all of those findings. If my opinion is incorrect, please revised the sentence.

Author Response

Notes to the comments of reviewer 2:

To comment 1:

There are a number of the mixed descriptions about Axl in HCC and other liver diseases, or in liver and other tissues. In addition, there are many redundant descriptions. The authors should clearly mention in each sentence what this description is related to. And the manuscript construction should be revised thoroughly.

Many thanks for the constructive criticism. We agree and substantially revised the manuscript construction by changing the order of chapters. We shifted chapter 3, which is now chapter 2 in the revised version, in order to bring the reader much faster to the main topic of Axl. We ask for the understanding of the reviewer that we cannot clearly mention in each sentence what this description is related to since such descriptions would end up in a high level of redundancy.

To comment 2:

Figure 1. Please add gamma-secretase and its cleaving site.

Thank you for the suggestion. There is only one publication showing the gamma-secretase-depending release of intracellular Axl. In addition, the cleavage site of the gamma-secretase is not mapped in the endodomain of Axl. As this review/manuscript focuses on ectodomain shedding of Axl which is demonstrated by multiple publications, we suggest to include only the ADAM10/17 cleavage site into Figure 1 rather than the less characterized cleavage site of gamma-secretase.

To comment 3:

The chapter 2 is not necessary because it is mainly focusing on other molecules than Axl, or should be revise by focusing on Axl.

In agreement with the reviewer, we revised chapter 2 (chapter 3 of the revised version) by rearranging statements, thus being more focused on Axl. In addition, we deleted the last two paragraphs dealing with ectodomain shedding and chemoresistance as this information is not related to our hypotheses how Axl signaling acts in liver fibrosis and HCC.

To comment 4:

How is the expression of Axl and Gas6 regulated in HCC? If such information is available, the authors should add it to the manuscript.

We appreciate this comment. We describe in chapter 3/part A (chapter 2/part A of the revised version) that HIF-1α, HIF-2α or RAB10 can transcriptionally activate Axl in HCC. Of note, no literature is available showing the regulation of Gas6 in HCC.

To comment 5:

Lines 214-237. How do TAM RTKs suppress inflammation in livers? If its molecular mechanism is revealed in detailed, it should be explained. What effects do they have on hepatitis?

We closely followed the suggestion of the reviewer. We inserted a new statement into chapter 3/part B (chapter 2/part B in the revised version) describing that the TAM triple knockout in mice results in hepatitis.

To comment 6:

The chapter 4. The authors should discuss about serum sAxl levels in HCC patients from the viewpoint of ADAM17 and/or gamma-secretase expression.

We agree with the request of the reviewer. No information on the ADAM17- and/or gamma-secretase-dependent release of sAxl is available in HCC patients after thoroughly screening the literature. Yet, increased ADAM10 expression correlates with increased Axl ectodomain shedding. Corresponding notions were inserted into chapter 4.

To comment 7:

Lines 299-300. This sentence is difficult to understand. I think that, even though sAxl suppresses the Gas6/Axl signaling, the excess amount of Gas6 compared to sAxl can explain all of those findings. If my opinion is incorrect, please revised the sentence.

Your opinion is correct. It is the excess amount of Gas6 compared to sAxl which could explain that the available Gas6 activates non-shedded Axl receptors for amplifying Gas6-dependent Axl signaling in liver fibrosis and HCC. We are convinced that this hypothesis is sufficiently explained in the text and in Figure 4B (right panel).

Reviewer 3 Report

I have reviewed the paper “Dynamics of Axl Receptor Shedding in

Hepatocellular Carcinoma and its Implication for Theranostics”. This is a well-written overview of an interesting subject. The authors managed to cover a broad spectrum of the field in a concise way and must be congratulated for this effort. This review has only several minor comments as follows.

1.    In the Section 5, authors described that the expression of AXL is associated with outcome after HCC resection. The location of the expression, i.e. tumor or the adjacent liver should be clarified because so-called “field effect” in the adjacent liver is also a strong predictor of HCC patients.

2.    Readers might want to know increases in sensitivity and specificity by adding serum sAxl to AFP.

3.    In line 361, authors described that the Axl expression is correlated to advanced stage and the tumor aggressiveness. If so, high Axl expression may be just a confounder or a bystander of poor outcomes. In the cited paper, the authors showed that the Axl expression was an independent factor by adjusting for the well-known clinical variables? If not, this description might be a little misleading.

Author Response

Notes to the comments of reviewer 3:

To comment 1:

In the Section 5, authors described that the expression of AXL is associated with outcome after HCC resection. The location of the expression, i.e. tumor or the adjacent liver should be clarified because so-called “field effect” in the adjacent liver is also a strong predictor of HCC patients.

Many thanks for the comment. Due to the lack of significant data, we deleted the entire paragraph dealing with Axl expression after HCC resection.

To comment 2:

Readers might want to know increases in sensitivity and specificity by adding serum sAxl to AFP.

Thank you for the suggestion. We inserted a notion into chapter 5 underlining that the combination of sAxl with AFP increases the diagnostic accuracy in very early to late HCC patients.

To comment 3:

In line 361, authors described that the Axl expression is correlated to advanced stage and the tumor aggressiveness. If so, high Axl expression may be just a confounder or a bystander of poor outcomes. In the cited paper, the authors showed that the Axl expression was an independent factor by adjusting for the well-known clinical variables? If not, this description might be a little misleading.

Many thanks for the comment. As already mentioned, we deleted the entire paragraph dealing with Axl expression after HCC resection due to the lack of significant data.

Round  2

Reviewer 2 Report

The authors sincerely and adequately responded to my comments. I have no further comments.